# Integrated Non-Factorized Variational Inference

**Shaobo Han**
Duke University
Durham, NC 27708
shaobo.han@duke.edu

**Xuejun Liao**
Duke University
Durham, NC 27708
xjliao@duke.edu

**Lawrence Carin**
Duke University
Durham, NC 27708
lcarin@duke.edu

## Abstract

We present a non-factorized variational method for full posterior inference in Bayesian hierarchical models, with the goal of capturing the posterior variable dependencies via efficient and possibly parallel computation. Our approach unifies the integrated nested Laplace approximation (INLA) under the variational framework. The proposed method is applicable in more challenging scenarios than typically assumed by INLA, such as Bayesian Lasso, which is characterized by the non-differentiability of the $\ell_1$ norm arising from independent Laplace priors. We derive an upper bound for the Kullback-Leibler divergence, which yields a fast closed-form solution via decoupled optimization. Our method is a reliable analytic alternative to Markov chain Monte Carlo (MCMC), and it results in a tighter evidence lower bound than that of mean-field variational Bayes (VB) method.

## 1 Introduction

Markov chain Monte Carlo (MCMC) methods [1] have been dominant tools for posterior analysis in Bayesian inference. Although MCMC can provide numerical representations of the exact posterior, they usually require intensive runs and are therefore time consuming. Moreover, assessment of a chain's convergence is a well-known challenge [2]. There have been many efforts dedicated to developing deterministic alternatives, including the Laplace approximation [3], variational methods [4], and expectation propagation (EP) [5]. These methods each have their merits and drawbacks [6].

More recently, the integrated nested Laplace approximation (INLA) [7] has emerged as an encouraging method for full posterior inference, which achieves computational accuracy and speed by taking advantage of a (typically) low-dimensional hyper-parameter space, to perform efficient numerical integration and parallel computation on a discrete grid. However, the Gaussian assumption for the latent process prevents INLA from being applied to more general models outside of the family of latent Gaussian models (LGMs).

In the machine learning community, variational inference has received significant use as an efficient alternative to MCMC. It is also attractive because it provides a closed-form lower bound to the model evidence. An active area of research has been focused on developing more efficient and accurate variational inference algorithms, for example, collapsed inference [8, 9], non-conjugate models [10, 11], multimodal posteriors [12], and fast convergent methods [13, 14].

The goal of this paper is to develop a reliable and efficient deterministic inference method, to both achieve the accuracy of MCMC and retain its inferential flexibility. We present a promising variational inference method without requiring the widely used factorized approximation to the posterior. Inspired by INLA, we propose a hybrid continuous-discrete variational approximation, which enables us to preserve full posterior dependencies and is therefore more accurate than the mean-field variational Bayes (VB) method [15]. The continuous variational approximation is flexible enough for various kinds of latent fields, which makes our method applicable to more general settings than assumed by INLA. The discretization of the low-dimensional hyper-parameter space can overcome the potential non-conjugacy and multimodal posterior problems in variational inference.

## 2 Integrated Non-Factorized Variational Bayesian Inference

Consider a general Bayesian hierarchical model with observation $\mathbf{y}$, latent variables $\mathbf{x}$, and hyperparameters $\boldsymbol{\theta}$. The exact joint posterior $p(\mathbf{x}, \boldsymbol{\theta}|\mathbf{y}) = p(\mathbf{y}, \mathbf{x}, \boldsymbol{\theta})/p(\mathbf{y})$ can be difficult to evaluate, since usually the normalization $p(\mathbf{y}) = \int \int p(\mathbf{y}, \mathbf{x}, \boldsymbol{\theta}) d\mathbf{x} d\boldsymbol{\theta}$ is intractable and numerical integration of $\mathbf{x}$ is too expensive.

To address this problem, we find a variational approximation to the exact posterior by minimizing the Kullback-Leibler (KL) divergence $\mathrm{KL}\left(q(\mathbf{x}, \boldsymbol{\theta}|\mathbf{y})||p(\mathbf{x}, \boldsymbol{\theta}|\mathbf{y})\right)$. Applying Jensen's inequality to the log-marginal data likelihood, one obtains

$$\ln p(\mathbf{y}) = \ln \int \int q(\mathbf{x}, \boldsymbol{\theta}|\mathbf{y}) \frac{p(\mathbf{y}, \mathbf{x}, \boldsymbol{\theta})}{q(\mathbf{x}, \boldsymbol{\theta}|\mathbf{y})} d\mathbf{x} d\boldsymbol{\theta} \geq \int \int q(\mathbf{x}, \boldsymbol{\theta}|\mathbf{y}) \ln \frac{p(\mathbf{y}, \mathbf{x}, \boldsymbol{\theta})}{q(\mathbf{x}, \boldsymbol{\theta}|\mathbf{y})} d\mathbf{x} d\boldsymbol{\theta} := \mathcal{L}, \quad (1)$$

which holds for any proposed approximating distributions $q(\mathbf{x}, \boldsymbol{\theta}|\mathbf{y})$. $\mathcal{L}$ is termed the evidence lower bound (ELBO)[4]. The gap in the Jensen's inequality is exactly the KL divergence. Therefore minimizing the Kullback-Leibler (KL) divergence is equivalent to maximizing the ELBO.

To make the variational problem tractable, the variational distribution $q(\mathbf{x}, \boldsymbol{\theta}|\mathbf{y})$ is commonly required to take a restricted form. For example, mean-field variational Bayes (VB) method assumes the distribution factorizes into a product of marginals [15], $q(\mathbf{x}, \boldsymbol{\theta}|\mathbf{y}) = q(\mathbf{x})q(\boldsymbol{\theta})$, which ignores the posterior dependencies among different latent variables (including hyperparameters) and therefore impairs the accuracy of the approximate posterior distribution.

### 2.1 Hybrid Continuous and Discrete Variational Approximations

We consider a non-factorized approximation to the posterior $q(\mathbf{x}, \boldsymbol{\theta}|\mathbf{y}) = q(\mathbf{x}|\mathbf{y}, \boldsymbol{\theta})q(\boldsymbol{\theta}|\mathbf{y})$, to preserve the posterior dependency structure. Unfortunately, this generally leads to a nontrivial optimization problem,

$$\begin{aligned} q^\star(\mathbf{x}, \boldsymbol{\theta}|\mathbf{y}) &= \operatorname{argmin}_{\{q(\mathbf{x}, \boldsymbol{\theta}|\mathbf{y})\}} \mathrm{KL}\left(q(\mathbf{x}, \boldsymbol{\theta}|\mathbf{y})||p(\mathbf{x}, \boldsymbol{\theta}|\mathbf{y})\right), \\ &= \operatorname{argmin}_{\{q(\mathbf{x}, \boldsymbol{\theta}|\mathbf{y})\}} \int \int q(\mathbf{x}, \boldsymbol{\theta}|\mathbf{y}) \ln \frac{q(\mathbf{x}, \boldsymbol{\theta}|\mathbf{y})}{p(\mathbf{x}, \boldsymbol{\theta}|\mathbf{y})} d\mathbf{x} d\boldsymbol{\theta}, \\ &= \operatorname{argmin}_{\{q(\mathbf{x}|\mathbf{y}, \boldsymbol{\theta}), q(\boldsymbol{\theta}|\mathbf{y})\}} \int q(\boldsymbol{\theta}|\mathbf{y}) \left[ \int q(\mathbf{x}|\boldsymbol{\theta}, \mathbf{y}) \ln \frac{q(\mathbf{x}|\boldsymbol{\theta}, \mathbf{y})}{p(\mathbf{x}, \boldsymbol{\theta}|\mathbf{y})} d\mathbf{x} + \ln q(\boldsymbol{\theta}|\mathbf{y}) \right] d\boldsymbol{\theta}. \quad (2) \end{aligned}$$

We propose a hybrid continuous-discrete variational distribution $q(\mathbf{x}|\mathbf{y}, \boldsymbol{\theta})q_d(\boldsymbol{\theta}|\mathbf{y})$, where $q_d(\boldsymbol{\theta}|\mathbf{y})$ is a finite mixture of Dirac-delta distributions, $q_d(\boldsymbol{\theta}|\mathbf{y}) = \sum_k \omega_k \delta_{\boldsymbol{\theta}_k}(\boldsymbol{\theta})$ with $\omega_k = q_d(\boldsymbol{\theta}_k|\mathbf{y})$ and $\sum_k \omega_k = 1$. Clearly, $q_d(\boldsymbol{\theta}|\mathbf{y})$ is an approximation of $q(\boldsymbol{\theta}|\mathbf{y})$ by discretizing the continuous (typically) low-dimensional parameter space of $\boldsymbol{\theta}$ using a grid $\mathcal{G}$ with finite grid points[1]. One can always reduce the discretization error by increasing the number of points in $\mathcal{G}$. To obtain a useful discretization at a manageable number of grid points, the dimension of $\boldsymbol{\theta}$ cannot be too large; this is also the same assumption in INLA [7], but we remove here the Gaussian prior assumption of INLA on latent effects $\mathbf{x}$.

The hybrid variational approximation is found by minimizing the KL divergence, i.e.,

$$\mathrm{KL}\left(q(\mathbf{x}, \boldsymbol{\theta}|\mathbf{y})||p(\mathbf{x}, \boldsymbol{\theta}|\mathbf{y})\right) = \sum_k q_d(\boldsymbol{\theta}_k|\mathbf{y}) \left[ \int q(\mathbf{x}|\boldsymbol{\theta}_k, \mathbf{y}) \ln \frac{q(\mathbf{x}|\mathbf{y}, \boldsymbol{\theta}_k)}{p(\mathbf{x}, \boldsymbol{\theta}_k|\mathbf{y})} d\mathbf{x} + \ln q_d(\boldsymbol{\theta}_k|\mathbf{y}) \right] \quad (3)$$

which leads to the approximate marginal posterior,

$$q(\mathbf{x}|\mathbf{y}) = \sum_k q(\mathbf{x}|\mathbf{y}, \boldsymbol{\theta}_k) q_d(\boldsymbol{\theta}_k|\mathbf{y}) \quad (4)$$

As will be clearer shortly, the problem in (3) can be much easier to solve than that in (2).

We give the name *integrated non-factorized variational Bayes* (INF-VB) to the method of approximating $p(\mathbf{x}, \boldsymbol{\theta}|\mathbf{y})$ with $q(\mathbf{x}|\mathbf{y}, \boldsymbol{\theta})q_d(\boldsymbol{\theta}|\mathbf{y})$ by solving the optimization problem in (3). The use of $q_d(\boldsymbol{\theta})$ is equivalent to numerical integration, which is a key idea of INLA [7], see Section 2.3 for details. It has also been used in sampling methods when samples are not easy to obtain directly [16]. Here we use this idea in variational inference to overcome the potential non-conjugacy and multimodal posterior problems in $\boldsymbol{\theta}$.

### 2.2 Variational Optimization

The proposed INF-VB method consists of two algorithmic steps:

- Step 1: Solving multiple independent optimization problems, each for a grid point in $\mathcal{G}$, to obtain the optimal $q(\mathbf{x}|\mathbf{y},\boldsymbol{\theta}_k)$, $\forall \boldsymbol{\theta}_k \in \mathcal{G}$, i.e.,

$$
\begin{aligned}
q^\star(\mathbf{x}|\mathbf{y},\boldsymbol{\theta}_k) &= \operatorname{argmin}_{\{q(\mathbf{x}|\mathbf{y},\boldsymbol{\theta}_k)\}} \sum_k q_d(\boldsymbol{\theta}_k|\mathbf{y}) \left[ \int q(\mathbf{x}|\boldsymbol{\theta}_k,\mathbf{y}) \ln \frac{q(\mathbf{x}|\mathbf{y},\boldsymbol{\theta}_k)}{p(\mathbf{x},\boldsymbol{\theta}_k|\mathbf{y})} d\mathbf{x} + \ln q_d(\boldsymbol{\theta}_k|\mathbf{y}) \right] \\
&= \operatorname{argmin}_{\{q(\mathbf{x}|\mathbf{y},\boldsymbol{\theta}_k)\}} \int q(\mathbf{x}|\boldsymbol{\theta}_k,\mathbf{y}) \ln \frac{q(\mathbf{x}|\mathbf{y},\boldsymbol{\theta}_k)}{p(\mathbf{x}|\mathbf{y},\boldsymbol{\theta}_k)} d\mathbf{x} \\
&= \operatorname{argmin}_{\{q(\mathbf{x}|\mathbf{y},\boldsymbol{\theta}_k)\}} \mathrm{KL}(q(\mathbf{x}|\mathbf{y},\boldsymbol{\theta}_k)\|p(\mathbf{x}|\mathbf{y},\boldsymbol{\theta}_k))
\end{aligned} \tag{5}
$$

  The optimal variational distribution $q^\star(\mathbf{x}|\mathbf{y},\boldsymbol{\theta}_k)$ is the exact posterior $p(\mathbf{x}|\mathbf{y},\boldsymbol{\theta}_k)$. In case it is not available, we may further constrain $q(\mathbf{x}|\mathbf{y},\boldsymbol{\theta}_k)$ to a parametric form, examples including: (i) multivariate Gaussian [17], if the posterior asymptotic normality holds; (ii) skew-normal densities [6, 18]; or (iii) an inducing factorization assumption (see Ch.10.2.5 in [19]), if the latent variables $\mathbf{x}$ are conditionally independent or their dependencies are negligible.

- Step 2: Given $\{q^\star(\mathbf{x}|\mathbf{y},\boldsymbol{\theta}_k) : \boldsymbol{\theta}_k \in \mathcal{G}\}$ obtained in Step 1, one solves

$$
\{q_d^\star(\boldsymbol{\theta}_k|\mathbf{y})\} = \operatorname{argmin}_{\{q_d(\boldsymbol{\theta}_k|\mathbf{y})\}} \sum_k q_d(\boldsymbol{\theta}_k|\mathbf{y}) \underbrace{\left[ \int q^\star(\mathbf{x}|\boldsymbol{\theta}_k,\mathbf{y}) \ln \frac{q^\star(\mathbf{x}|\mathbf{y},\boldsymbol{\theta}_k)}{p(\mathbf{x},\boldsymbol{\theta}_k|\mathbf{y})} d\mathbf{x} + \ln q_d(\boldsymbol{\theta}_k|\mathbf{y}) \right]}_{l(q_d(\boldsymbol{\theta}_k|\mathbf{y}))=l(\omega_k)}
$$

  Setting $\partial l(\omega_k)/\partial \omega_k = 0$ (also $\partial^2 l(\omega_k)/\partial \omega_k^2 > 0$), which is solved to give

$$
q_d^\star(\boldsymbol{\theta}_k|\mathbf{y}) \propto \exp\left( \int q^\star(\mathbf{x}|\mathbf{y},\boldsymbol{\theta}_k) \ln \frac{p(\mathbf{x},\boldsymbol{\theta}_k|\mathbf{y})}{q^\star(\mathbf{x}|\mathbf{y},\boldsymbol{\theta}_k)} d\mathbf{x} \right). \tag{6}
$$

  Note that $q_d(\boldsymbol{\theta}|\mathbf{y})$ is evaluated at a grid of points $\boldsymbol{\theta}_k \in \mathcal{G}$, it needs to be known only up to a multiplicative constant, which can be identified from the normalization constraint $\sum_k q_d^\star(\boldsymbol{\theta}_k|\mathbf{y}) = 1$. The integral in (6) can be analytically evaluated in the application considered in Section 3.

## 2.3 Links between INF-VB and INLA

The INF-VB is a variational extension of the integrated nested Laplace approximations (INLA) [7], a deterministic Bayesian inference method for latent Gaussian models (LGMs), to the case when $p(\mathbf{x}|\boldsymbol{\theta})$ exhibits strong non-Gaussianity and hence $p(\boldsymbol{\theta}|\mathbf{y})$ may not be approximated accurately by the Laplace's method of integration [20]. To see the connection, we review briefly the three computation steps of INLA and compare them with INF-VB in below:

1. Based on the Laplace approximation [3], INLA seeks a Gaussian distribution $q_G(\mathbf{x}|\mathbf{y},\boldsymbol{\theta}_k) = \mathcal{N}(\mathbf{x};\mathbf{x}^*(\boldsymbol{\theta}_k),\mathbf{H}(\mathbf{x}^*(\boldsymbol{\theta}_k))^{-1})$, $\forall \boldsymbol{\theta}_k \in \mathcal{G}$ that captures most of the probabilistic mass locally, where $\mathbf{x}^*(\boldsymbol{\theta}_k) = \operatorname{argmax}_{\mathbf{x}} p(\mathbf{x}|\mathbf{y},\boldsymbol{\theta}_k)$ is the posterior mode, and $\mathbf{H}(\mathbf{x}^*(\boldsymbol{\theta}_k))$ is the Hessian matrix of the log posterior evaluated at the mode. By contrast, INF-VB with the Gaussian parametric constraint on $q^\star(\mathbf{x}|\mathbf{y},\boldsymbol{\theta}_k)$ provides a global variational Gaussian approximation $q_{VG}(\mathbf{x}|\mathbf{y},\boldsymbol{\theta}_k)$ in the sense that the conditions of the Laplace approximation hold on average [17]. As we will see next, the averaging operator plays a crucial role in handling the non-differentiable $\ell_1$ norm arising from the double-exponential priors.

2. INLA computes the marginal posteriors of $\boldsymbol{\theta}$ based on the Laplace's method of integration [20],

$$
q_{LA}(\boldsymbol{\theta}|\mathbf{y}) = \left. \frac{p(\mathbf{x},\boldsymbol{\theta}|\mathbf{y})}{q(\mathbf{x}|\mathbf{y},\boldsymbol{\theta})} \right|_{\mathbf{x}=\mathbf{x}^*(\boldsymbol{\theta})} \tag{7}
$$

  The quality of this approximation depends on the accuracy of $q(\mathbf{x}|\mathbf{y},\boldsymbol{\theta})$. When $q(\mathbf{x}|\mathbf{y},\boldsymbol{\theta}) = p(\mathbf{x}|\mathbf{y},\boldsymbol{\theta})$, one has $q_{LA}(\boldsymbol{\theta}|\mathbf{y})$ equal to $p(\boldsymbol{\theta}|\mathbf{y})$, according to the Bayes rule. It has been shown in [7] that (7) is accurate enough for latent Gaussian models with $q_G(\mathbf{x}|\mathbf{y},\boldsymbol{\theta})$. Alternatively, the variational optimal posterior $q_d^\star(\boldsymbol{\theta}|\mathbf{y})$ by INF-VB (6) can be derived as a lower bound of the true posterior $p(\boldsymbol{\theta}|\mathbf{y})$ by Jensen's inequality.

$$
\ln p(\boldsymbol{\theta}|\mathbf{y}) = \ln\left[ \int \frac{p(\mathbf{x},\boldsymbol{\theta}|\mathbf{y})}{q(\mathbf{x}|\mathbf{y},\boldsymbol{\theta})} q(\mathbf{x}|\mathbf{y},\boldsymbol{\theta}) d\mathbf{x} \right] \geq \int \ln\left[ \frac{p(\mathbf{x},\boldsymbol{\theta}|\mathbf{y})}{q(\mathbf{x}|\mathbf{y},\boldsymbol{\theta})} q(\mathbf{x}|\mathbf{y},\boldsymbol{\theta}) \right] d\mathbf{x} = \ln q_d^\star(\boldsymbol{\theta}|\mathbf{y}) \tag{8}
$$

  Its optimality justifications in Section 2.2 also explain the often observed empirical successes of hyperparameter selection based on the ELBO of $\ln p(\mathbf{y}|\boldsymbol{\theta})$ [13], when the first level of Bayesian inference is performed, i.e. only the conditional posterior $q(\mathbf{x}|\mathbf{y},\boldsymbol{\theta})$ with fixed $\boldsymbol{\theta}$ is of interest. In Section 4 we compare the accuracies of both (6) and (7) for hyperparameter learning.

3. INLA obtains the marginal distributions of interest, e.g., $q(\mathbf{x}|\mathbf{y})$ via numerically integrating out $\boldsymbol{\theta}$: $q(\mathbf{x}|\mathbf{y}) = \sum_k q(\mathbf{x}|\mathbf{y},\boldsymbol{\theta}_k)q(\boldsymbol{\theta}_k|\mathbf{y})\boldsymbol{\Delta}_k$ with area weights $\boldsymbol{\Delta}_k$. In INF-VB, we have $q_d(\boldsymbol{\theta}|\mathbf{y}) = \sum_k \omega_k \delta_{\boldsymbol{\theta}_k}(\boldsymbol{\theta})$. Let $\omega_k = q(\boldsymbol{\theta}_k|\mathbf{y})\boldsymbol{\Delta}_k$, we immediately have

$$q(\mathbf{x}|\mathbf{y}) = \int q(\mathbf{x}|\mathbf{y},\boldsymbol{\theta})q_d(\boldsymbol{\theta}|\mathbf{y})d\boldsymbol{\theta} = \sum_k q(\mathbf{x}|\mathbf{y},\boldsymbol{\theta}_k)q_d(\boldsymbol{\theta}_k|\mathbf{y}) = \sum_k q(\mathbf{x}|\mathbf{y},\boldsymbol{\theta}_k)q(\boldsymbol{\theta}_k|\mathbf{y})\boldsymbol{\Delta}_k \quad (9)$$

This Dirac-delta mixture interpretation of numerical integration also enables us to quantitize the accuracy of INLA approximation $q_G(\mathbf{x}|\mathbf{y},\boldsymbol{\theta})q_{LA}(\boldsymbol{\theta}|\mathbf{y})$ using the KL divergence to $p(\mathbf{x},\boldsymbol{\theta}|\mathbf{y})$ under the variational framework.

In contrast to INLA, INF-VB provides $q(\mathbf{x}|\mathbf{y},\boldsymbol{\theta})$ and $q_d(\boldsymbol{\theta}|\mathbf{y})$, both are optimal in a sense of the minimum Kullback-Leibler divergence, within the proposed hybrid distribution family. In this paper we focus on the full posterior inference of Bayesian Lasso [21] where the local Laplace approximation in INLA cannot be applied, as the non-differentiability of the $\ell_1$ norm prevents one from computing the Hessian matrix. Besides, if we do not exploit the scale mixture of normals representation [22] of Laplace priors (i.e., no data-augmentation), we are actually dealing with a non-conjugate variational inference problem in Bayesian Lasso.

## 3   Application to Bayesian Lasso

Consider the Bayesian Lasso regression model [21], $\mathbf{y} = \boldsymbol{\Phi}\mathbf{x} + \mathbf{e}$, where $\boldsymbol{\Phi} \in \mathbb{R}^{n \times p}$ is the design matrix containing predictors, $\mathbf{y} \in \mathbb{R}^n$ are responses[2], and $\mathbf{e} \in \mathbb{R}^n$ contain independent zero-mean Gaussian noise $\mathbf{e} \sim \mathcal{N}(\mathbf{e}; \mathbf{0}, \sigma^2\mathbf{I}_n)$. Following [21] we assume[3],

$$x_j|\sigma^2, \lambda^2, \sim \frac{\lambda}{2\sqrt{\sigma^2}}\exp\left(-\frac{\lambda}{\sqrt{\sigma^2}}\|x_j\|_1\right), \quad \sigma^2 \sim \text{InvGamma}(\sigma^2; a, b), \quad \lambda^2 \sim \text{Gamma}(\lambda^2; r, s)$$

While the Lasso estimates [23] provide only the posterior modes of the regression parameters $\mathbf{x} \in \mathbb{R}^p$, Bayesian Lasso [21] provides the complete posterior distribution $p(\mathbf{x}, \boldsymbol{\theta}|\mathbf{y})$, from which one may obtain whatever statistical properties are desired of $\mathbf{x}$ and $\boldsymbol{\theta}$, including the posterior mode, mean, median, and credible intervals.

Since in our approach variational Gaussian approximation is performed separately (see Section 3.1) for each hyperparameter $\{\lambda, \sigma^2\}$ considered, the efficiency of approximating $p(\mathbf{x}|\mathbf{y}, \boldsymbol{\theta})$ is particularly important. The upper bound of the KL divergence derived in Section 3.2 provides an approximate closed-form solution, that is often accurate enough or requires a small number of gradient iterations to converge to optimality. The tightness of the upper bound is analyzed using spectral-norm bounds (See Section 3.3), which also provide insights on the connection between the deterministic Lasso [23] and the Bayesian Lasso [21].

### 3.1   Variational Gaussian Approximation

The conditional distribution of $\mathbf{y}$ and $\mathbf{x}$ given $\boldsymbol{\theta}$ is

$$p(\mathbf{y}, \mathbf{x}|\boldsymbol{\theta}) = \frac{\lambda^p/(2\sigma)^p}{\sqrt{(2\pi\sigma^2)^n}}\exp\left\{-\frac{\|\mathbf{y}-\boldsymbol{\Phi}\mathbf{x}\|^2}{2\sigma^2} - \frac{\lambda}{\sigma}\|\mathbf{x}\|_1\right\}. \quad (10)$$

The postulated approximation, $q(\mathbf{x}|\boldsymbol{\theta},\mathbf{y}) = \mathcal{N}(\mathbf{x}; \boldsymbol{\mu}, \mathbf{D})$, is a multivariate Gaussian density (dropping dependencies of variational parameters $(\boldsymbol{\mu}, \mathbf{D})$ on $(\boldsymbol{\theta}, \mathbf{y})$ for brevity), whose parameters $(\boldsymbol{\mu}, \mathbf{D})$ are found by minimizing the KL divergence to $p(\mathbf{x}|\boldsymbol{\theta}, \mathbf{y})$,

$$g(\boldsymbol{\mu}, \mathbf{D}) \overset{Def.}{=} \text{KL}(q(\mathbf{x}; \boldsymbol{\mu}, \mathbf{D})\|p(\mathbf{x}|\mathbf{y}, \boldsymbol{\theta})) = \int q(\mathbf{x}; \boldsymbol{\mu}, \mathbf{D}) \ln\frac{q(\mathbf{x};\boldsymbol{\mu},\mathbf{D})}{p(\mathbf{x}|\mathbf{y},\boldsymbol{\theta})}d\mathbf{x}$$

$$= \int q(\mathbf{x}; \boldsymbol{\mu}, \mathbf{D}) \ln\frac{q(\mathbf{x};\boldsymbol{\mu},\mathbf{D})}{p(\mathbf{y},\mathbf{x}|\boldsymbol{\theta})}d\mathbf{x} + \ln p(\mathbf{y}|\boldsymbol{\theta}),$$

$$= -\frac{1}{2}\ln|\mathbf{D}| + \frac{\|\mathbf{y}-\boldsymbol{\Phi}\boldsymbol{\mu}\|^2 + \text{tr}(\boldsymbol{\Phi}'\boldsymbol{\Phi}\mathbf{D})}{2\sigma^2} + \frac{\lambda}{\sigma}\mathbb{E}_q(\|\mathbf{x}\|_1) + \ln p(\mathbf{y}|\boldsymbol{\theta}) - \ln\psi(\sigma^2, \lambda)$$

$$\mathbb{E}_q(\|\mathbf{x}\|_1) = \sum_{j=1}^p\left[\mu_j - 2\mu_j\Psi(h_j) + 2\sqrt{d_j}\psi(h_j)\right], \quad h_j = -\mu_j\sqrt{d_j}, \quad d_j = \mathbf{D}_{jj} \quad (11)$$

where $\psi(\sigma^2, \lambda) = (4\pi e\lambda^2\sigma^{-2})^{p/2}(2\pi\sigma^2)^{-n/2}$, $\Psi(\cdot)$ and $\psi(\cdot)$ corresponds to the standard normal cumulative distribution function and probability density function, respectively. Expectation is taken with respect to $q(\mathbf{x}; \boldsymbol{\mu}, \mathbf{D})$. Define $\mathbf{D} = \mathbf{C}\mathbf{C}^T$, where $\mathbf{C}$ is the Cholesky factorization of the covariance matrix $\mathbf{D}$. Since $g(\boldsymbol{\mu}, \mathbf{D})$ is convex in the parameter space $(\boldsymbol{\mu}, \mathbf{C})$, a global optimal variational Gaussian approximation $q^\star(\mathbf{x}|\mathbf{y}, \boldsymbol{\theta})$ is guaranteed, which achieves the minimum KL divergence to $p(\mathbf{x}|\boldsymbol{\theta}, \mathbf{y})$ within the family of multivariate Gaussian densities specified [13][4].

As a first step, one finds $q^\star(\mathbf{x}|\mathbf{y}, \boldsymbol{\theta})$ using gradient based procedures independently for each hyperparameter combinations $\{\lambda, \sigma^2\}$. Second, $q^\star(\boldsymbol{\theta}|y)$ can be evaluated analytically using either (6) or (7); both will yield a finite mixture of Gaussian distribution for the marginal posterior $q(\mathbf{x}|\mathbf{y})$ via numerical integration, which is highly efficient since we only have two hyperparameters in Bayesian Lasso. Finally, the evidence lower bound (ELBO) in (1) can also be evaluated analytically after simple algebra. We will show in Section 4.3 a comparison with the mean-field variational Bayesian (VB) approach, derived based on a scale normal mixture representation [22] of the Laplace prior.

## 3.2 Upper Bounds of KL divergence

We provide an approximate solution $(\hat{\boldsymbol{\mu}}, \hat{\mathbf{D}})$ via minimizing an upper bound of KL divergence (11). This solution solves a Lasso problem in $\boldsymbol{\mu}$, and has a closed-form expression for $\mathbf{D}$, making this computationally efficient. In practice, it could serve as an initialization for gradient procedures.

**Lemma 3.1.** *(Triangle Inequality)* $\mathbb{E}_q\|\mathbf{x}\|_1 \leq \mathbb{E}_q\|\mathbf{x} - \boldsymbol{\mu}\|_1 + \|\boldsymbol{\mu}\|_1$, *where* $\mathbb{E}_q\|\mathbf{x} - \boldsymbol{\mu}\|_1 = \sqrt{2/\pi} \sum_{j=1}^p \sqrt{\mathbf{d}_j}$, *with the expectation taken with respect to* $q(\mathbf{x}; \boldsymbol{\mu}, \mathbf{D})$.

**Lemma 3.2.** *For any* $\{d_j \geq 0\}_{j=1}^p$, *it holds* $\sqrt{\sum_{j=1}^p d_j^2} \leq \sum_{j=1}^p d_j \leq \sqrt{p \sum_{j=1}^p d_j^2}$.

**Lemma 3.3.** *[24] For any* $\mathbf{A} \in \mathbb{S}_{++}^p$, $\mathrm{tr}(\mathbf{A}^2) \leq \mathrm{tr}(\mathbf{A}) \leq \sqrt{p}\,\mathrm{tr}(\mathbf{A}^2)$.

**Theorem 3.1.** *(Upper and Lower bound) For any* $\mathbf{A}, \mathbf{D} \in \mathbb{S}_{++}^p$, $\mathbf{A} = \sqrt{\mathbf{D}}$[5], $d_j = \mathbf{D}_{jj}$ *holds* $\frac{1}{\sqrt{p}}\mathrm{tr}(\mathbf{A}) \leq \sum_{j=1}^p \sqrt{d_j} \leq \sqrt{p}\,\mathrm{tr}(\mathbf{A})$.

Applying Lemma 3.1 and Theorem 3.1 in (11), one obtains an upper bound for KL divergence,

$$f(\boldsymbol{\mu}, \mathbf{D}) = \underbrace{\frac{\|\mathbf{y} - \boldsymbol{\Phi}\boldsymbol{\mu}\|_2^2}{2\sigma^2} + \frac{\lambda}{\sigma}\|\boldsymbol{\mu}\|_1}_{f_1(\boldsymbol{\mu})} + \underbrace{-\frac{1}{2}\ln|\mathbf{D}| + \frac{\mathrm{tr}(\boldsymbol{\Phi}'\boldsymbol{\Phi}\mathbf{D})}{2\sigma^2} + \frac{\lambda}{\sigma}\sqrt{\frac{2p}{\pi}}\mathrm{tr}(\sqrt{\mathbf{D}})}_{f_2(\mathbf{D})} + \ln\frac{p(\mathbf{y}|\boldsymbol{\theta})}{\psi(\sigma^2, \lambda)}$$

$$\geq g(\boldsymbol{\mu}, \mathbf{D}) = \mathrm{KL}(q(\mathbf{x}; \boldsymbol{\mu}, \mathbf{D})\|p(\mathbf{x}|\mathbf{y}, \boldsymbol{\theta})) \tag{12}$$

In the problem of minimizing the KL divergence $g(\boldsymbol{\mu}, \mathbf{CC}^T)$, one needs to iteratively update $\boldsymbol{\mu}$ and $\mathbf{C}$, since they are coupled. However, the upper bound $f(\boldsymbol{\mu}, \mathbf{D})$ decouples into two additive terms: $f_1$ is a function of $\boldsymbol{\mu}$ while $f_2$ is a function of $\mathbf{D}$, which greatly simplifies the minimization.

- The minimization of $f_1(\boldsymbol{\mu})$ is a convex Lasso problem. Using path-following algorithms (e.g., a modified least angle regression algorithm (LARS) [25]), one can efficiently compute the entire solution path of Lasso estimates as a function of $\lambda_0 = 2\lambda\sigma$ in one shot. Global optimal solutions for $\hat{\boldsymbol{\mu}}(\boldsymbol{\theta}_k)$ on each grid point $\boldsymbol{\theta}_k \in \mathcal{G}$ can be recovered using the piece-wise linear property.

- The function $f_2(\mathbf{D})$ is convex in the parameter space $\mathbf{A} = \sqrt{\mathbf{D}}$, whose minimizer is in closed-form and can be found by setting the gradient to zero and solving the resulting equation,

$$\nabla_{\mathbf{A}} f_2 = -\mathbf{A}^{-1} + \frac{\boldsymbol{\Phi}'\boldsymbol{\Phi}\mathbf{A}}{\sigma^2} + \lambda\sqrt{\frac{2p}{\pi}}\mathbf{I} = 0, \quad \hat{\mathbf{A}} = \left(\sqrt{\frac{\lambda^2 p}{2\pi\sigma^2}}\mathbf{I} + \sqrt{\frac{\lambda^2 p}{2\pi\sigma^2}\mathbf{I} + \frac{\boldsymbol{\Phi}'\boldsymbol{\Phi}}{\sigma^2}}\right)^{-1}, \tag{13}$$

We have $\hat{\mathbf{D}} = \hat{\mathbf{A}}^2$, which is guaranteed to be a positive definite matrix. Note that the global optimum $\hat{\mathbf{D}}(\boldsymbol{\theta}_k)$ for each grid point $\boldsymbol{\theta}_k \in \mathcal{G}$ have the same eigenvectors as the Gram matrix $\boldsymbol{\Phi}'\boldsymbol{\Phi}$ and differ only in eigenvalues. For $j = 1, \dots, p$, denote the eigenvalues of $\mathbf{D}$ and $\boldsymbol{\Phi}'\boldsymbol{\Phi}$ as $\alpha_j$ and $\beta_j$, respectively. By (13), we have $\alpha_j = \lambda\sqrt{p/(2\pi\sigma^2)} + \sqrt{\lambda^2 p/(2\pi\sigma^2) + \beta_j/\sigma^2}$. Therefore, one can pre-compute the eigenvectors once, and only update the eigenvalues as a function of $\boldsymbol{\theta}_k$. This will make the computation efficient both in time and memory.

The solutions $(\hat{\boldsymbol{\mu}}, \hat{\mathbf{D}})$ which minimize the KL upper bound $f(\hat{\boldsymbol{\mu}}, \hat{\mathbf{D}})$ in (12) achieves its global optimum. Meanwhile, it is also accurate in the sense of the KL divergence $g(\hat{\boldsymbol{\mu}}, \hat{\mathbf{D}})$ in (11), as we will show next. Tightness analysis of the upper bound is also provided, using trace norm bounds.

### 3.3 Theoretical Anlaysis

**Theorem 3.2.** *(KL Divergence Upper Bound) Let* $(\hat{\boldsymbol{\mu}}, \hat{\mathbf{D}})$ *be the minimizer of the KL upper bound(12), i.e.,* $\hat{\boldsymbol{\mu}}$ *solves the Lasso and* $\hat{\mathbf{D}}$ *is given in (13). Then*

$$g(\hat{\boldsymbol{\mu}}, \hat{\mathbf{D}}) \leq \min_{\boldsymbol{\mu}, \mathbf{D}} f(\boldsymbol{\mu}, \mathbf{D}) = f_1(\hat{\boldsymbol{\mu}}) + f_2(\hat{\mathbf{D}}) + \ln \frac{p(\mathbf{y}|\boldsymbol{\theta})}{\psi(\sigma^2, \lambda)} \qquad (14)$$

*where* $f_1(\hat{\boldsymbol{\mu}}) = \min_{\boldsymbol{\mu}} \left( \frac{\|\mathbf{y} - \boldsymbol{\Phi}\boldsymbol{\mu}\|_2^2}{2\sigma^2} + \frac{\lambda}{\sigma} \|\boldsymbol{\mu}\|_1 \right)$, $f_2(\hat{\mathbf{D}}) = \sum_j \ln \alpha_j + \sum_j \frac{\beta_j \alpha_j^{-2}}{2\sigma^2} + \sum_j \sqrt{\frac{2\lambda^2 n}{\pi}} (\alpha_j)^{-1}$.
Thus the KL divergence for $(\hat{\boldsymbol{\mu}}, \hat{\mathbf{D}})$ is upper bounded by the minimum achievable $\ell_1$-penalized least square error $\epsilon_1 = f_1(\hat{\boldsymbol{\mu}})$ and terms in $f_2(\hat{\mathbf{D}})$ which are ultimately related to the eigenvalues $\{\beta_j\}$ $(j = 1, \dots, p)$ of the Gram matrix $\boldsymbol{\Phi}'\boldsymbol{\Phi}$.

Let $(\boldsymbol{\mu}^*, \mathbf{D}^*)$ be the minimizer of the original KL divergence $g(\boldsymbol{\mu}, \mathbf{D})$, and $g_1(\boldsymbol{\mu}|\mathbf{D})$ collect the terms of $g(\boldsymbol{\mu}, \mathbf{D})$ that are related to $\boldsymbol{\mu}$. Then the Bayesian posterior mean obtained via VG, i.e.,

$$\boldsymbol{\mu}^* = \text{argmin}_{\boldsymbol{\mu}} \, g_1(\boldsymbol{\mu}|\mathbf{D}^*) = \text{argmin}_{\boldsymbol{\mu}} \, \mathbb{E}_{q(\mathbf{x}|\mathbf{y}, \boldsymbol{\theta})} \left( \|\mathbf{y} - \boldsymbol{\Phi}\mathbf{x}\|_2^2 + 2\lambda\sigma\|\mathbf{x}\|_1 \right), \qquad (15)$$

is a counterpart of the deterministic Lasso [23], which appears naturally in the upper bound,

$$\hat{\boldsymbol{\mu}} = \text{argmin}_{\boldsymbol{\mu}} \, f_1(\boldsymbol{\mu}) = \text{argmin}_{\boldsymbol{\mu}} \left( \|\mathbf{y} - \boldsymbol{\Phi}\boldsymbol{\mu}\|_2^2 + 2\lambda\sigma\|\boldsymbol{\mu}\|_1 \right) \qquad (16)$$

Note that the Lasso solution cannot be found by gradient methods due to non-differentiability. By taking the expectation, the objective function is smoothed around $\mathbf{0}$ and thus differentiable. This connection indicates that in VG for Bayesian Lasso, the conditions of deterministic Lasso hold on average, with respect to the variational distribution $q(\mathbf{x}|\mathbf{y}, \boldsymbol{\theta})$, in the parameter space of $\boldsymbol{\mu}$.

The following theorem (proof sketches are in the Supplementary Material) provides quantitative measures of the closeness of the upper bounds, $f_1(\boldsymbol{\mu})$ and $f(\boldsymbol{\mu}, \mathbf{D})$, to their respective true counterparts.

**Theorem 3.3.** *The tightness of* $f_1(\boldsymbol{\mu})$ *and* $f(\boldsymbol{\mu}, \mathbf{D})$ *is given by*

$$g_1(\boldsymbol{\mu}|\mathbf{D}) - f_1(\boldsymbol{\mu}) \leq \frac{\text{tr}(\boldsymbol{\Phi}'\boldsymbol{\Phi}\mathbf{D})}{2\sigma^2} + \frac{\lambda}{\sigma}\sqrt{\frac{2p}{\pi}}\text{tr}(\sqrt{\mathbf{D}}), \quad f(\boldsymbol{\mu}, \mathbf{D}) - g(\boldsymbol{\mu}, \mathbf{D}) \leq \frac{2\lambda}{\sigma}\sqrt{\frac{2p}{\pi}}\text{tr}(\sqrt{\mathbf{D}}) \, (17)$$

*which holds for any* $(\boldsymbol{\mu}, \mathbf{D}) \in \mathbb{R}^p \times \mathbb{S}_{++}^p$. *Further assume* $g(\boldsymbol{\mu}^*, \mathbf{D}^*) = \epsilon_2$ *(minimum achievable KL divergence, or information gap), we have*

$$f_1(\boldsymbol{\mu}^*) \leq g_1(\boldsymbol{\mu}^*) \leq g_1(\hat{\boldsymbol{\mu}}) \leq \epsilon_1 + \text{tr}(\boldsymbol{\Phi}'\boldsymbol{\Phi}\mathbf{D})/(2\sigma^2) + \lambda\sqrt{2p/(\sigma^2\pi)}\text{tr}(\sqrt{\hat{\mathbf{D}}}) \qquad (18a)$$

$$g(\hat{\boldsymbol{\mu}}, \hat{\mathbf{D}}) \leq f(\hat{\boldsymbol{\mu}}, \hat{\mathbf{D}}) \leq f(\boldsymbol{\mu}^*, \mathbf{D}^*) \leq \epsilon_2 + 2\lambda\sqrt{2p/(\sigma^2\pi)}\text{tr}(\sqrt{\mathbf{D}^*}) \qquad (18b)$$

## 4 Experiments

We consider long runs of **MCMC** [6] as reference solutions, and consider two types of INF-VB: **INF-VB-1** calculates hyperparameter posteriors using (6); while **INF-VB-2** uses (7) and evaluates it at the posterior mode of $p(\mathbf{x}|\mathbf{y}, \boldsymbol{\theta})$. We also compare INF-VB-1 and INF-VB-2 to **VB**, a mean-field variational Bayes (VB) solution (See Supplementary Material for update equations). The results show that the INF-VB method is more accurate than VB, and is a promising alternative to MCMC for Bayesian Lasso.

### 4.1 Synthetic Dataset

We compare the proposed INF-VB methods with VB and intensive MCMC runs, in terms of the joint posterior $q(\lambda^2, \sigma^2|\mathbf{y})$, the marginal posteriors of hyper-parameters $q(\sigma^2|\mathbf{y})$ and $q(\lambda^2|\mathbf{y})$, and the marginal posteriors of regression coefficients $q(x_j|\mathbf{y})$ (see Figure 1). The observations are generated from $y_i = \boldsymbol{\phi}_i^T \mathbf{x} + \epsilon_i$, $i = 1, \dots, 600$, where $\phi_{ij}$ are drawn from an *i.i.d.* normal distribution[7], where the pairwise correlation between the $j$th and the $k$th columns of $\boldsymbol{\Phi}$ is $0.5^{|j-k|}$; we draw $\epsilon_i \sim \mathcal{N}(0, \sigma^2)$, $x_j|\lambda, \sigma \sim \text{Laplace}(\lambda/\sigma)$, $j = 1, \dots, 300$, and set $\sigma^2 = 0.5$, $\lambda = 0.5$.

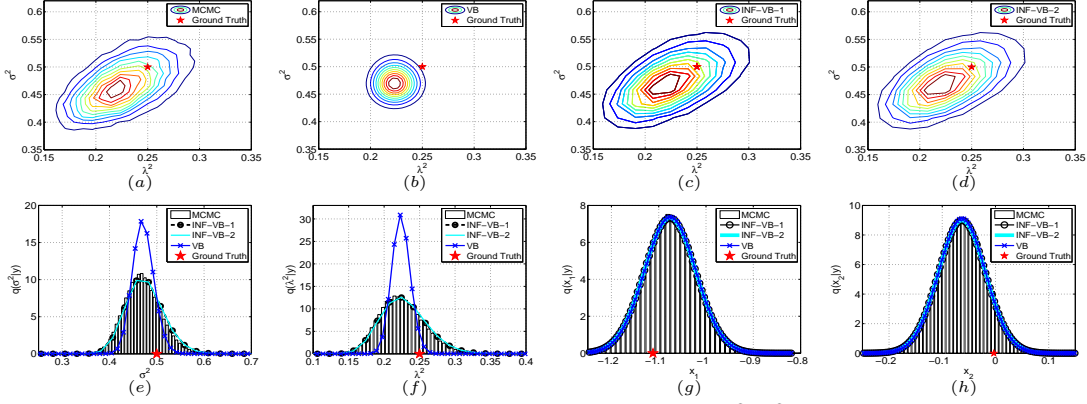

Figure 1: Contour plots for joint posteriors of hyperparameters $q(\sigma^2, \lambda^2|\mathbf{y})$: (a)-(d); Marginal posterior of hyperparameters and coefficients: $(e)$ $q(\sigma^2|\mathbf{y})$, (f)$q(\lambda^2|\mathbf{y})$; (g) $q(x_1|\mathbf{y})$, (h)$q(x_2|\mathbf{y})$

See Figure 1(a)-(d), both MCMC and INF-VB preserve the strong posterior dependence among hyperparameters, while mean-field VB cannot. While mean-field VB approximates the posterior mode well, the posterior variance can be (sometimes severely) underestimated, see Figure 1$(e)$, $(f)$. Since we have analytically approximated $p(\mathbf{x}|\mathbf{y})$ by a finite mixture of normal distribution $q(\mathbf{x}|\mathbf{y}, \boldsymbol{\theta})$ with mixing weights $q(\boldsymbol{\theta}|\mathbf{y})$, the posterior marginals for the latent variables: $q(x_j|\mathbf{y})$ are easily accessible from this analytical representation. Perhaps surprisingly, both INF-VB and mean-field VB provide quite accurate marginal distributions $q(x_j|\mathbf{y})$, see Figure 1(j)-(h) for examples. The differences in the tails of $q(\boldsymbol{\theta}|\mathbf{y})$ between INF-VB and mean-field VB yield negligible differences in the marginal distributions $q(x_j|\mathbf{y})$, when $\boldsymbol{\theta}$ is integrated out.

## 4.2 Diabetes Dataset

We consider the benchmark diabetes dataset [25] frequently used in previous studies of Bayesian Lasso; see [21, 26], for example. The goal of this diagnostic study, as suggested in [25], is to construct a linear regression model ($n = 442$, $p = 10$) to reveal the important determinants of the response, and to provide interpretable results to guide disease progression. In Figure 2, we show accurate marginal posteriors of hyperparameters $q(\sigma^2|\mathbf{y})$ and $q(\lambda^2|\mathbf{y})$ as well as marginals of coefficients $q(x_j|\mathbf{y})$, $j = 1, \ldots, 10$, which indicate the relevance of each predictor. We also compared them to the ordinary least square (OLS) estimates.

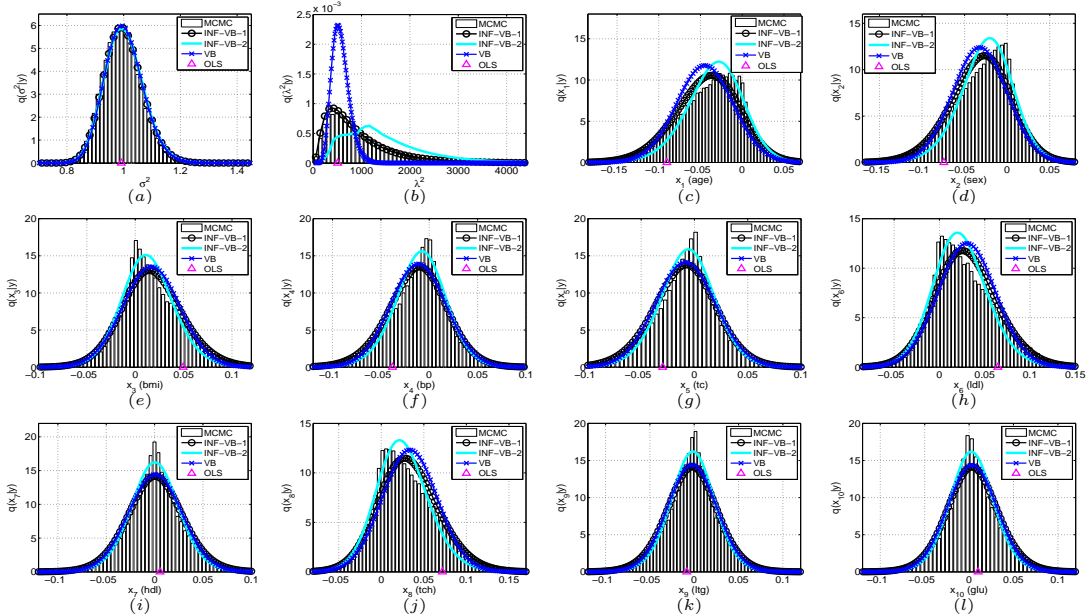

Figure 2: Posterior marginals of hyperparameters: (a) $q(\sigma^2|\mathbf{y})$ and (b)$q(\lambda^2|\mathbf{y})$; posterior marginals of coefficients: (c)-(l) $q(x_j|\mathbf{y})$ ($j = 1, \ldots, 10$)

## 4.3 Comparison: Accuracy and Speed

We quantitatively measure the quality of the approximate joint probability $q(\mathbf{x}, \boldsymbol{\theta}|\mathbf{y})$ provided by our non-factorized variational methods, and compare them to VB under factorization assumptions. The KL divergence $\mathrm{KL}(q(\mathbf{x}, \boldsymbol{\theta}|\mathbf{y})|p(\mathbf{x}, \boldsymbol{\theta}|\mathbf{y}))$ is not directly available; instead, we compare the negative evidence lower bound (1), which can be evaluated analytically in our case and differs from the KL divergence only up to a constant. We also measure the computational time of different algorithms by elapsed times (seconds). In INF-VB, different grids of sizes $m \times m$ are considered, where $m = 1, 5, 10, 30, 50$. We consider two real world datasets: the above Diabetes dataset, and the Prostate cancer dataset [27]. Here, **INF-VB-3** and **INF-VB-4** refer to the methods that use the approximate solution in Section 3.2 with no gradient steps for $q(\mathbf{x}|\mathbf{y}, \boldsymbol{\theta})$, and use (6) or (7) for $q(\boldsymbol{\theta}|\mathbf{y})$.

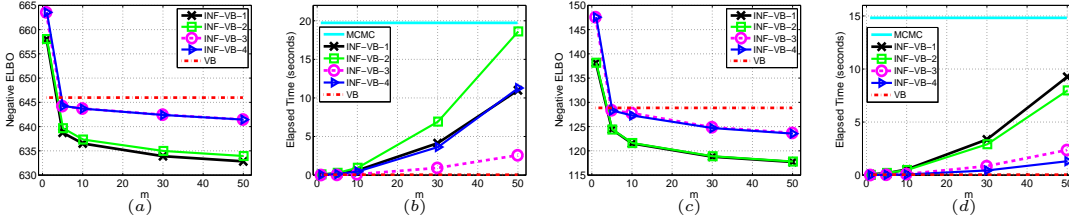

Figure 3: Negative evidence lower bound (ELBO) and elapsed time v.s. grid size; (a), (b) for the Diabetes dataset ($n = 442$, $p = 10$). (c), (d) for the Prostate cancer dataset ($n = 97$, $p = 8$)

The quality of variational methods depends on the flexibility of variational distributions. In INF-VB for Bayesian Lasso, we constrain $q(\mathbf{x}|\mathbf{y}, \boldsymbol{\theta})$ to be parametric and $q(\boldsymbol{\theta}|\mathbf{y})$ to be still in free form. See from Figure 3, the accuracy of INF-VB method with a $1 \times 1$ grid is worse than mean-field VB, which corresponds to the partial Bayesian learning of $q(\mathbf{x}|\mathbf{y}, \boldsymbol{\theta})$ with a fixed $\boldsymbol{\theta}$. As the grid size increases, the accuracies of INF-VB (even those without gradient steps) also increase and are in general of better quality than mean-field VB, in the sense of negative ELBO (KL divergence up to a constant).

The computational complexities of INF-VB, mean-field VB, and MCMC methods are proportional to the grid size, number of iterations toward local optimum, and the number of runs, respectively. Since the computations on the grid are independent, INF-VB is highly parallelizable, which is an important feature as more multiprocessor computational power becomes available. Besides, one may further reduce its computational load by choosing grid points more economically, which will be pursued in our next step. Even the small datasets we show here for illustration enjoy good speed-ups. A significant speed-up for INF-VB can be achieved via parallel computing.

## 5 Discussion

We have provided a flexible framework for approximate inference of the full posterior $p(\mathbf{x}, \boldsymbol{\theta}|\mathbf{y})$ based on a hybrid continuous-discrete variational distribution, which is optimal in the sense of the KL divergence. As a reliable and efficient alternative to MCMC, our method generalizes INLA to non-Gaussian priors and VB to non-factorization settings. While we have used Bayesian Lasso as an example, our inference method is generically applicable. One can also approximate $p(\mathbf{x}|\mathbf{y}, \boldsymbol{\theta})$ using other methods, such as scalable variational methods [28], or improved EP [29].

The posterior $p(\boldsymbol{\theta}|\mathbf{y})$, which is analyzed based on a grid approximation, enables users to do both model averaging and model selection, depending on specific purposes. The discretized approximation of $p(\boldsymbol{\theta}|\mathbf{y})$ overcomes the potential non-conjugacy or multimodal issues in the $\boldsymbol{\theta}$ space in variational inference, and it also allows parallel implementation of the hybrid continuous-discrete variational approximation with the dominant computational load (approximating the continuous high dimensional $q(\mathbf{x}|\mathbf{y}, \boldsymbol{\theta})$) distributed on each grid point, which is particularly important when applying INF-VB to large-scale Bayesian inference. INF-VB has limitations. The number of hyperparameters $\boldsymbol{\theta}$ should be no more than 5 to 6, which is the same fundamental limitation of INLA.

### Acknowledgments

The work reported here was supported in part by grants from ARO, DARPA, DOE, NGA and ONR.

## Footnotes

[1] The grid points need not to be uniformly spaced, one may put more grid points to potentially high mass regions if credible prior information is available.

[2]We assume that both $\mathbf{y}$ and the columns of $\boldsymbol{\Phi}$ have been mean-centered to remove the intercept term.

[3][21] suggested using scaled double-exponential priors under which they showed that $p(\mathbf{x}, \sigma^2|\mathbf{y}, \lambda)$ is unimodal, further, the unimodality helps to accelerate convergence of the data-augmentation Gibbs sampler and makes the posterior mode more meaningful. Gamma prior is put on $\lambda^2$ for conjugacy.

[4]Code for variational Gaussian approximation is available at mloss.org/software/view/308

[5]Since $\mathbf{D}$ is positive definite, it has a unique symmetric square root $\mathbf{A} = \sqrt{\mathbf{D}}$, which can be obtained from $\mathbf{D}$ by taking square root of the eigenvalues.

[6]In all experiments shown here, we take intensive MCMC runs as the gold standard (with $5 \times 10^3$ burn-ins and $5 \times 10^5$ samples collected). We use data-augmentation Gibbs sampler introduced in [21]. Ground truth for latent variables and hyper-parameter are also compared to whenever possible. The hyperparameters for Gamma distributions are set to $a = b = r = s = 0.001$ through all these experiments. If not mentioned, the grid size is $50 \times 50$, which is uniformly created around the ordinary least square (OLS) estimates of hyper-parameters.

[7]The responses $\mathbf{y}$ and the columns of $\boldsymbol{\Phi}$ are centered; the columns of $\boldsymbol{\Phi}$ are also scaled to have unit variance

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
