[Supplementary Material]

# Integrated Non-Factorized Variational Inference
# Supplementary Material

**Shaobo Han**
Duke University
Durham, NC 27708
shaobo.han@duke.edu

**Xuejun Liao**
Duke University
Durham, NC 27708
xjliao@duke.edu

**Lawrence Carin**
Duke University
Durham, NC 27708
lcarin@duke.edu

**Proof of Theorem 3.1**

Each inequality is obtained by first applying Lemma 3.2 and then Lemma 3.3,

$$\sum_{j=1}^{p} \sqrt{d_j} \leq \sqrt{p \sum_{j=1}^{p} \sqrt{d_j}} = \sqrt{p}\,\mathrm{tr}(\mathbf{D}) = \sqrt{p}\,\mathrm{tr}(\mathbf{A}^2) \leq \sqrt{p}\,\mathrm{tr}(\mathbf{A}) = \sqrt{p}\,\mathrm{tr}(\sqrt{\mathbf{D}}),$$

$$\sum_{j=1}^{p} \sqrt{d_j} \geq \sqrt{\sum_{j=1}^{p} \sqrt{d_j}} = \mathrm{tr}(\mathbf{D}) = \mathrm{tr}(\mathbf{A}^2) \geq \frac{1}{\sqrt{p}}\mathrm{tr}(\mathbf{A}) = \frac{1}{\sqrt{p}}\mathrm{tr}(\sqrt{\mathbf{D}}).$$

Theorem 3.2 holds according to the upper bound of KL divergence and the proof is straightforward.

**Proof of Theorem 3.3**

To see the first inequality in (17), we have

$$
\begin{aligned}
g_1(\boldsymbol{\mu}) - f_1(\boldsymbol{\mu}) &= \frac{\mathrm{tr}(\boldsymbol{\Phi}'\boldsymbol{\Phi}\mathbf{D})}{2\sigma^2} + \frac{\lambda}{\sigma}\mathbb{E}_q(\|\mathbf{x}\|_1 - \|\boldsymbol{\mu}\|_1) \leq \frac{\mathrm{tr}(\boldsymbol{\Phi}'\boldsymbol{\Phi}\mathbf{D})}{2\sigma^2} + \frac{\lambda}{\sigma}\mathbb{E}_q(\|\mathbf{x} - \boldsymbol{\mu}\|_1) \\
&= \frac{\mathrm{tr}(\boldsymbol{\Phi}'\boldsymbol{\Phi}\mathbf{D})}{2\sigma^2} + \frac{\lambda}{\sigma}\sqrt{\frac{2}{\pi}}\sum_j \sqrt{d_j} \leq \frac{\mathrm{tr}(\boldsymbol{\Phi}'\boldsymbol{\Phi}\mathbf{D})}{2\sigma^2} + \frac{\lambda}{\sigma}\sqrt{\frac{2p}{\pi}}\mathrm{tr}(\sqrt{\mathbf{D}})
\end{aligned}
$$

holds for any $\boldsymbol{\mu} \in \mathbb{R}^p$. Note that $f(\boldsymbol{\mu}, \mathbf{D})$ is an upper bound of $g(\boldsymbol{\mu}, \mathbf{D})$, $f_1(\boldsymbol{\mu}^*) \leq g_1(\boldsymbol{\mu}^*)$ (the proof is straightforward). Thus the first inequality in (18a) holds . The second inequality holds since $\boldsymbol{\mu}^*$ is the global minimum of $g_1(\boldsymbol{\mu})$. To see the second inequality in (17), we have that

$$
\begin{aligned}
f(\boldsymbol{\mu}, \mathbf{D}) - g(\boldsymbol{\mu}, \mathbf{D}) &= \frac{\lambda}{\sigma}\sqrt{\frac{2p}{\pi}}\mathrm{tr}(\sqrt{\mathbf{D}}) + \frac{\lambda}{\sigma}\mathbb{E}_q\left(\|\boldsymbol{\mu}\|_1 - \|\mathbf{x}\|_1\right) \leq \frac{\lambda}{\sigma}\sqrt{\frac{2p}{\pi}}\mathrm{tr}(\sqrt{\mathbf{D}}) + \frac{\lambda}{\sigma}\mathbb{E}_q\left(\|\boldsymbol{\mu} - \mathbf{x}\|_1\right) \\
&= \frac{\lambda}{\sigma}\sqrt{\frac{2p}{\pi}}\mathrm{tr}(\sqrt{\mathbf{D}}) + \frac{\lambda}{\sigma}\sqrt{\frac{2}{\pi}}\sum_j \sqrt{d_j} \leq 2\frac{\lambda}{\sigma}\sqrt{\frac{2p}{\pi}}\mathrm{tr}(\sqrt{\mathbf{D}})
\end{aligned}
$$

holds for any $(\boldsymbol{\mu}, \mathbf{D}) \in \mathbb{R}^p \times \mathbb{S}_{++}^p$. The first inequality in (18b) holds since $f(\boldsymbol{\mu}, \mathbf{D})$ is a upper bound of $g(\boldsymbol{\mu}, \mathbf{D})$; the second inequality holds since $(\hat{\boldsymbol{\mu}}, \hat{\mathbf{D}})$ is the global minimum of $f(\boldsymbol{\mu}, \mathbf{D})$.

**Bayesian Lasso Model (Scaled Case)**

According to [1], the Bayesian Lasso model with the scale-mixture of normal representation is as follows,

$$
\begin{aligned}
\mathbf{y}|\mathbf{x}, \sigma^2 &\sim \mathcal{N}_n(\mathbf{y}; \boldsymbol{\Phi}\mathbf{x}, \sigma^2\mathbf{I}_n) \\
\mathbf{x}|\sigma^2, \tau_1^2, \ldots, \tau_p^2 &\sim \mathcal{N}_p(\mathbf{x}; \mathbf{0}_p, \sigma^2\mathbf{D}_{\boldsymbol{\tau}}), \quad \mathbf{D}_{\boldsymbol{\tau}} = \mathrm{diag}(\tau_1^2, \ldots, \tau_p^2) \\
\tau_1^2, \ldots, \tau_p^2 &\sim \prod_{j=1}^{p} \frac{\lambda^2}{2}\exp\left(-\lambda^2\tau_j^2/2\right)d\tau_j^2, \quad \tau_1^2, \ldots, \tau_p^2 > 0, \quad j = 1, \ldots, p \\
\gamma_j &\sim \frac{\lambda^2}{2}\exp\left(-\lambda^2/2\gamma_j\right)\gamma_j^{-2}, \quad \gamma_j = 1/\tau_j^2, \quad j = 1, \ldots, p
\end{aligned}
$$

$$\sigma^2 \sim \text{InvGamma}(\sigma^2; a, b)$$
$$\lambda^2 \sim \text{Gamma}(\lambda^2; r, s) \qquad (1)$$

where representation of Laplace distribution as a scale mixture of normals (with an exponential mixing density) is exploited,

$$\frac{t}{2} \exp(-t|z|) = \int_0^\infty \frac{1}{\sqrt{2\pi s}} \exp(-z^2/(2s)) \frac{t^2}{2} \exp(-t^2 s/2) ds \qquad (2)$$

where $t > 0$, $t = \lambda/\sigma$, $s = \sigma^2 \tau_j^2 = \sigma^2/\gamma_j$.

## Data-Augmentation Gibbs Sampler

The full likelihood can be written as follows,

$$p(\mathbf{y}|\mathbf{x}, \sigma^2) p(\mathbf{x}|\sigma^2, \boldsymbol{\gamma}) p(\boldsymbol{\gamma}) p(\sigma^2) p(\lambda^2) = \mathcal{N}_n(\mathbf{y}; \boldsymbol{\Phi}\mathbf{x}, \sigma^2 \mathbf{I}_n) \mathcal{N}_p(\mathbf{x}; \mathbf{0}_p, \sigma^2 \mathbf{D}_\tau) \left( \prod_{i=1}^p p(\gamma_j) \right) p(\sigma^2) p(\lambda)$$

$$= \frac{1}{(2\pi)^{n/2}|\sigma^2 \mathbf{I}_n|^{1/2}} \exp\left( -\frac{(\mathbf{y} - \boldsymbol{\Phi}\mathbf{x})^T(\mathbf{y} - \boldsymbol{\Phi}\mathbf{x})}{2\sigma^2} \right) \times \frac{1}{(2\pi)^{p/2}[\prod_{j=1}^p \sigma^2/\gamma_j]^{1/2}} \exp\left( -\frac{\mathbf{x}^T \mathbf{D}_\tau^{-1}\mathbf{x}}{2\sigma^2} \right)$$

$$\times \prod_{j=1}^p \left( \frac{\lambda^2}{2} \exp(-\lambda^2/(2\gamma_j))(\gamma_j^{-2}) \right) \times \frac{b^a}{\Gamma(a)}(\sigma^2)^{-(a+1)} \exp(-b/\sigma^2) \times \frac{s^r}{\Gamma(r)} \lambda^{2(r-1)} \exp(-s\lambda^2) \quad (3)$$

where $\mathbf{D}_\tau = \text{diag}(1/\gamma_1, \ldots, 1/\gamma_p)$.

- Full Conditional distribution of $\mathbf{x}$:
$$(\mathbf{x}|\mathbf{y}, \sigma^2, \tau_1^2, \ldots, \tau_p^2) \sim \mathcal{N}_p(\mathbf{x}; (\mathbf{D}_\tau^{-1} + \boldsymbol{\Phi}^T\boldsymbol{\Phi})^{-1}\boldsymbol{\Phi}^T\mathbf{y}, \sigma^2(\mathbf{D}_\tau^{-1} + \boldsymbol{\Phi}^T\boldsymbol{\Phi})^{-1}) \qquad (4)$$

- Full Conditional distribution of $\sigma^2$:
$$(\sigma^2|\mathbf{y}, \tau_1^2, \ldots, \tau_p^2) \sim \text{InvGamma}(\sigma^2; \frac{n+p-1}{2} + a, \frac{(\mathbf{y} - \boldsymbol{\Phi}\mathbf{x})^T(\mathbf{y} - \boldsymbol{\Phi}\mathbf{x}) + \mathbf{x}^T\mathbf{D}_\tau^{-1}\mathbf{x}}{2} + b) \quad (5)$$

- Full Conditional distribution of $\gamma_j = 1/\tau_j^2$:
$$p(1/\tau_j^2|\lambda^2, \sigma^2, x_j) \propto (1/\tau_j^2)^{-\frac{3}{2}} \exp\{-\left( \frac{(x_j/\tau_j^2 - \lambda\sigma)^2}{2\sigma^2(1/\tau_j^2)} \right)\} \sim \text{InvGaussian}(1/\tau_j^2; g, h) \qquad (6)$$

  where $g = \sqrt{\lambda^2\sigma^2/x_j^2}$ and $h = \lambda^2$.

- Full Conditional distribution of $\lambda^2$:
$$(\lambda^2|\tau_j^2) \sim \text{Gamma}(\lambda^2; p + r, s + \sum_{j=1}^p \frac{\tau_j^2}{2}) \qquad (7)$$

## Mean-field VB

We seek a variational distribution $q(\boldsymbol{\Theta}; \boldsymbol{\Gamma})$ to approximate the exact posterior $p(\boldsymbol{\Theta}; \boldsymbol{\Gamma})$, where $\boldsymbol{\Theta} \equiv \{\mathbf{x}, \boldsymbol{\gamma}, \sigma^2, \lambda^2\}$, $\boldsymbol{\Gamma}$ are the variational parameters. Consider the variational expression,

$$\tilde{F}(\boldsymbol{\Gamma}) = \int d\boldsymbol{\Theta} q(\boldsymbol{\Theta}; \boldsymbol{\Gamma}) \ln \frac{q(\boldsymbol{\Theta}; \boldsymbol{\Gamma})}{p(\mathbf{y})p(\boldsymbol{\Theta}|\mathbf{y})} = -\ln p(\mathbf{y}) + \text{KL}(q(\boldsymbol{\Theta}; \boldsymbol{\Gamma})||p(\boldsymbol{\Theta}|\mathbf{y})) \qquad (8)$$

Note that the term $p(\mathbf{y})$ is a constant with respect to $\boldsymbol{\Gamma}$, and therefore the evidence lower bound $\tilde{F}(\boldsymbol{\Gamma})$ is maximized when the Kullback-Leibler divergence $\text{KL}(q(\boldsymbol{\Theta}; \boldsymbol{\Gamma})||p(\boldsymbol{\Theta}|\mathbf{y}))$ is minimized. To make the computation of $\tilde{F}(\boldsymbol{\Gamma})$ tractable, we assume $q(\boldsymbol{\Theta}; \boldsymbol{\Gamma})$ has a factorized form,

$$q(\boldsymbol{\Theta}; \boldsymbol{\Gamma}) = \prod_{i=1}^k q_i(\Theta_i; \Gamma_i) \qquad (9)$$

With appropriate choice of $q_i$, the variational expression $\tilde{F}(\boldsymbol{\Gamma})$ may be evaluated analytically. Maximizing the lower bound $\tilde{F}(\boldsymbol{\Gamma})$ with respect to $q_i^\star(\Theta_i; \Gamma_i)$ yields

$$q_i^\star(\Theta_i; \Gamma_i) = \frac{\exp(\mathbb{E}_{i\neq j}[\ln p(\mathbf{y}, \boldsymbol{\Theta})])}{\int \exp(\mathbb{E}_{i\neq j}[\ln p(\mathbf{y}, \boldsymbol{\Theta})]) d\Theta_i} \qquad (10)$$

The update equations are as follows,

- Update for $\mathbf{x}$:
$$q^\star(\mathbf{x}|-) \sim \mathcal{N}(\mathbf{x}; \hat{\boldsymbol{\mu}}, \hat{\boldsymbol{\Sigma}})$$

$$\hat{\boldsymbol{\mu}} \;=\; \left(\langle \mathbf{D}_{\boldsymbol{\tau}}^{-1}\rangle + \boldsymbol{\Phi}^T\boldsymbol{\Phi}\right)^{-1}\boldsymbol{\Phi}^T\mathbf{y}, \quad \hat{\boldsymbol{\Sigma}} = \left[\langle\sigma^{-2}\rangle\left(\langle\mathbf{D}_{\boldsymbol{\tau}}^{-1}\rangle + \boldsymbol{\Phi}^T\boldsymbol{\Phi}\right)\right]^{-1} \quad (11)$$

where $\langle\sigma^{-2}\rangle = \hat{a}/\hat{b}$

- Update for $\sigma^{-2}$:

$$
\begin{aligned}
q^\star(\sigma^{-2}|-) &\sim \text{Gamma}(\sigma^{-2};\hat{a},\hat{b}) \\
\hat{a} &= \frac{n+p-1}{2} + a \\
\hat{b} &= \frac{1}{2}\mathbf{y}^T\mathbf{y} - \mathbf{y}^T\boldsymbol{\Phi}\langle\mathbf{x}\rangle + \frac{1}{2}\text{trace}\left[\left(\boldsymbol{\Phi}^T\boldsymbol{\Phi} + \langle\mathbf{D}_{\boldsymbol{\tau}}^{-1}\rangle\right)\langle\mathbf{x}\mathbf{x}^T\rangle\right] + b \quad (12)
\end{aligned}
$$

where $\langle\mathbf{x}\rangle = \hat{\boldsymbol{\mu}}$, $\langle\mathbf{x}\mathbf{x}^T\rangle = \hat{\boldsymbol{\mu}}\hat{\boldsymbol{\mu}}^T + \hat{\boldsymbol{\Sigma}}$.

- Update for $\lambda^2$:

$$q^\star(\lambda^2|-) \sim \text{Gamma}(\lambda^2;\hat{r},\hat{s}), \quad \hat{r} = p+r, \quad \hat{s} = \sum_{j=1}^p \langle\frac{1}{2\gamma_j}\rangle + s \quad (13)$$

- Update for $\boldsymbol{\gamma}_j, j=1,...,p$:

$$q^\star(\gamma_j|-) \sim \text{InvGaussian}(\gamma_j;\hat{g}_j,\hat{h}_j), \quad \hat{g}_j = \sqrt{\frac{\langle\lambda^2\rangle}{\langle\sigma^{-2}\rangle\langle x_j^2\rangle}}, \quad \hat{h}_j = \langle\lambda^2\rangle \quad (14)$$

where $\text{InvGaussian}(x;g,h) = \sqrt{\frac{h}{2\pi x^3}}\exp\left(-\frac{h(x-g)^2}{2g^2 x}\right)$

$(x>0)$ denotes the inverse Gaussian distribution with mean $\langle x\rangle = g$ and $\langle x^{-1}\rangle = g^{-1} + h^{-1}$. We have

$$\langle\lambda^2\rangle = \hat{r}/\hat{s}, \quad \langle x_j^2\rangle = \hat{\mu}_j^2 + \hat{\Sigma}_{jj}, \quad \langle\gamma_j^{-1}\rangle = \hat{g}_j^{-1} + \hat{h}_j^{-1}, \quad \langle\mathbf{D}_{\boldsymbol{\tau}}^{-1}\rangle = \text{diag}\left[\hat{g}_j\right]_{j=1:p} \quad (15)$$

The lower bound $\tilde{F}(\boldsymbol{\Gamma})$ can be calculated very straightforwardly both for tracking the monotonic increase and for possibly setting a convergence criterion.

$$
\begin{aligned}
\tilde{F}(\boldsymbol{\Gamma}) &= \langle\ln p(\mathbf{y}|\mathbf{x},\sigma^2)\rangle + \langle\ln p(\mathbf{x}|\sigma^2,\boldsymbol{\gamma})\rangle + \langle\ln p(\boldsymbol{\gamma})\rangle + \langle\ln p(\sigma^2)\rangle + \langle\ln p(\lambda^2)\rangle \\
&\quad - \langle\ln q^\star(\mathbf{x}|-)\rangle - \langle\ln q^\star(\boldsymbol{\gamma}|-)\rangle - \langle\ln q^\star(\sigma^{-2}|-)\rangle - \langle\ln q^\star(\lambda^2|-)\rangle \quad (16)
\end{aligned}
$$

where

$$
\begin{aligned}
\langle\ln p(\mathbf{y}|\mathbf{x},\sigma^2)\rangle &= -\frac{n}{2}\ln 2\pi + \frac{n}{2}\langle\ln\sigma^{-2}\rangle - \frac{1}{2}\langle\sigma^{-2}\rangle\left(||\mathbf{y}-\boldsymbol{\Phi}\hat{\boldsymbol{\mu}}||_2^2 + \text{trace}(\boldsymbol{\Phi}^T\boldsymbol{\Phi}\hat{\boldsymbol{\Sigma}})\right) \\
\langle\ln\sigma^{-2}\rangle &= \psi(\hat{a}) - \ln(\hat{b}), \quad \langle\sigma^{-2}\rangle = \hat{a}/\hat{b} \quad (17)
\end{aligned}
$$

and $\psi(\cdot)$ is the digamma function.

$$\langle\ln p(\mathbf{x}|\sigma^2,\boldsymbol{\gamma})\rangle = -\frac{p}{2}\ln 2\pi + \frac{p}{2}\langle\ln\sigma^{-2}\rangle + \frac{1}{2}\sum_{j=1}^p\langle\ln\gamma_j\rangle - \frac{1}{2}\langle\sigma^{-2}\rangle\text{trace}\left(\langle\mathbf{D}_{\boldsymbol{\tau}}^{-1}\rangle\langle\mathbf{x}\mathbf{x}^T\rangle\right) \quad (18)$$

and those $\langle\ln\gamma_j\rangle$ terms canceled out.

$$
\begin{aligned}
\langle\ln p(\boldsymbol{\gamma})\rangle &= \sum_{j=1}^p\langle\log p(\gamma_j)\rangle = \sum_{j=1}^p\left(\langle\ln\frac{\lambda^2}{2}\rangle - \langle\gamma_j^{-1}\rangle\langle\frac{\lambda^2}{2}\rangle - 2\langle\ln\gamma_j\rangle\right) \\
\langle\ln\lambda^2\rangle &= \psi(\hat{r}) - \ln(\hat{s}) \quad (19)
\end{aligned}
$$

$$\langle\ln p(\sigma^{-2})\rangle = \ln\left(\frac{b^a}{\Gamma(a)}\right) + (a-1)\langle\ln\sigma^{-2}\rangle - b\langle\sigma^{-2}\rangle \quad (20)$$

$$\langle\ln p(\lambda^2)\rangle = \ln\left(\frac{s^r}{\Gamma(r)}\right) + (r-1)\langle\ln\lambda^2\rangle - s\langle\lambda^2\rangle \quad (21)$$

$$-\langle\ln q^\star(\mathbf{x}|-)\rangle = \frac{1}{2}\ln|2\pi e\hat{\boldsymbol{\Sigma}}| \quad (22)$$

$$-\langle\ln q^\star(\boldsymbol{\gamma}|-)\rangle = -\sum_{j=1}^p\langle\ln q^\star(\gamma_j|-)\rangle = \sum_{j=1}^p\left(-\frac{1}{2}\ln\hat{h}_j + \frac{1}{2}\ln 2\pi + \frac{3}{2}\langle\ln\gamma_j\rangle + 0.5\right) \quad (23)$$

$$-\langle\ln q^\star(\sigma^{-2}|-)\rangle = -\hat{a}\ln\hat{b} + \ln\Gamma(\hat{a}) - (\hat{a}-1)\langle\ln\sigma^{-2}\rangle + \hat{b}\langle\sigma^{-2}\rangle \quad (24)$$

$$-\langle\ln q^\star(\lambda^2|-)\rangle = -\hat{r}\ln\hat{s} + \ln\Gamma(\hat{r}) - (\hat{r}-1)\langle\ln\lambda^2\rangle + \hat{s}\langle\lambda^2\rangle \quad (25)$$

# References

[1]  T. Park and G. Casella. The Bayesian Lasso. *J. Am. Statist. Assoc.*, 103(482):681–686, 2008.