[Reviews · NeurIPS 2013]

Submitted by Assigned_Reviewer_5

This paper proposed Integrated Non-Factorized Variational (INF-VB) inference.
The INF-VB inference aims at solving the problem of the factorization assumption of the VB inference and includes the integrated nested Laplace approximation (INLA) as a special case.
Therefore, the proposed method is applicable to more general settings than assumed by INLA.
The theory and empirical evaluation is convincing.
A weak point is that this framework is limited to the continuous variational approximation as well as INLA.
Summary: A novel non-factorized variational Bayes inference is proposed.
The theoretical analysis and the empirical evaluation is convincing.

Submitted by Assigned_Reviewer_7

Overview
==
The paper proposed a form of variational distribution where dependency is assumed in the approximation. We have q(x,\theta) = q(\theta)q(x|\theta), and q(\theta) is represented as a series of Dirac-deltas. The idea resembles INLA, but with the clear advantages of VB (i.e. a bound on the marginal likelihood.)


Discussion
==
Introduction
--
This is a good introduction which lays out the paper and makes me curious the read further.

Section 2
--
The ideas seems like a good one, and you describe the method in some detail in this section. The problem boils down the the minimisation of the objective in (2).

My concern here is the entropy of the 'gridded' parameter. You move from a continuous integral in (2) to a sum with Dirac deltas in (3): what happens to the entropy in q(\theta)?

2.3:
I'm not an expert on INLA, so perhaps other reviewers can comment here. I like the idea of a comparison, but struggle a little with your motivation. You satae that INLA cannot cope with sever non-Gaussianity in the posterior: how do you propose that the VB method will work better? I understand that the approximation is different, and that the variational method is based on a well-defined metric (KL divergence), but I fail to see how it helps with Gaussianity. Perhaps just a change of language is needed?

Section 3
--
Here you lay out the application of the method to Bayesian Lasso. The idea is to marginalise the parameters \theta (which I guess are the noise variance \sigma^2 and the prior width \lambda) on a grid, and for each grid point compute the optimal distribution q(x|\theta). It's great that the optimiation of each conditional distribution is convex: could you comment on wht would happen if it were not? Would it be necessary to find the global optimimum each time, or would the method be robust to that?

On the first paragraph of page 5, you describe how the method preceeds, but I'm still concerned about the entropy of q(\theta).

3.2 and 3.3:
The upper bound on the KL divergence is an interesting proposition. I guess it's specific to the Bayesian Lasso model since it relies on the triangle inequality in Lemma 3.1.

I confess that I'm a bit bamboozled by the idea of a Baysesian Lasso. To me, the lasso is a shrinkage method which helps in selecting a sparse point-estimnate solution. The Bayesian posterior is necessarily not sparse. Perhaps I should read the Bayesian Lasso paper, but you might include some more description of the model?

Section 4
--
Very nice experiments, well presented results which make the advantages clear. It troubles me a little that these experiments are rather small though, it would have been nice to see it applied to bigger data sets.


Pros
==
- A nice formal framework, with (theoretical) comparison to INLA
- A relevant and interesting problem, of interest to large parts of the community
- an interesting upper bound on the KL which gives a one-shot solution (or initialisation)

Cons
==
- Application is presented only to the Bayesian Lasso
- Experiments are well presented but a little limited

Summary
==
An interesting idea which is of interest to large parts of the community, with just enough experimental evidence to convey its practicality.

Errata
==
160: integrat_ing_ out.


Summary: A variational version of INLA. Some points to clarify but otherwise a nice paper. Experiments are a bit small, but forgivable.

Submitted by Assigned_Reviewer_8

This paper proposes an approximation to the variational posterior over hidden variables and parameters in models where the hidden variables are Gaussian. The approximation consists of discretizing the parameter space and optimizing the weights for the discrete values. The theoretical analysis for the Bayesian Lasso case is interesting, and the experiments are simple but thorough. I wonder about the narrow applicability to models with low-dim parameter spaces, perhaps in high-dim cases the parameters can be factorized and each subspace would be discretized separately.

Quality: the paper is technically sound.
Clarity: the paper is well written, organized, and easy to follow.
Originality: discretizing a hidden subspace is a rather straightforward idea (though carrying out the optimization procedure is nontrivial).
Significance: the range of applications may be limited to models interesting more to the statistics than the machine learning community.


Summary: A well-written paper proposing a simple scheme for approximating the joint variables-parameters posterior in hidden Gaussian models using a variational approach.
Author Feedback

Author rebuttal: We thank the reviewers for spending time reading our manuscript and providing their thoughtful comments. Our responses are provided below.

> To Review 1:

Q: A weak point is that this framework is limited to the continuous variational approximation as well as INLA.

A: We infer x and \theta, both assumed continuous. In the discrete case, the inference may be simpler. For example, if \theta is discrete by nature, the approximation q_d(\theta_k|y) is unnecessary. We will examine this in future work. Thank you for pointing this out.

> To Reviewer 2:

Q1: What happens to the entropy in q(\theta)?

A1: Excellent question. The differential entropy of q(\theta|y) is approximated by the discrete entropy of q_d(\theta_k|y), k=1,2,.... According to Theorem 8.3.1 of (Cover and Thomas, 2006), the difference between the differential entropy and the discrete entropy is approximated by log(\Delta), where \Delta describes the area of each cell in a uniform grid. In our case, the difference is a constant, since \Delta is fixed.


Q2: How do you propose that the VB method will work better?
... how it helps with Gaussianity. Perhaps just a change of language is needed?

A2: INFVB provides a global Gaussian approximation for p(x|y,\theta), which is better than the local Laplace approximation of INLA in the sense of minimum KL distance. Besides, INFVB is applicable to more general cases including severely non-Gaussian posteriors. For example, one may use a skew-normal approximation when the underlying posterior p(x|y,\theta) is skewed.

We will improve the language for these and enhance the description of Bayesian lasso in the final paper. Thank you for your suggestions.


Q3: Application is presented only to the Bayesian Lasso.

A3: Both our approach and INLA are applicable to a broad class of models in statistical applications, including (generalized) linear models and additive models. One drawback of the original INLA is its inability to handle sparseness-promoting priors (e.g., Bayesian Lasso), which are highly important in machine learning applications. We focus on sparseness-promoting priors in this paper. Since this is an important new aspect of the proposed method, we have chosen to simplify the presentation by only considering a linear model for the likelihood. It is relatively straightforward to extend what we present here to a generalized linear model, such as logistic regression.


> To Reviewer 3:

Thank you for your comments.